# Sub-synaptic, multiplexed analysis of proteins reveals Fragile X related protein 2 is mislocalized in *Fmr1* KO synapses

Gordon X Wang[1,2,3]*, Stephen J Smith[3,5], Philippe Mourrain[1,2,4]

[1]Department of Psychiatry and Behavioral Sciences, Stanford University School of Medicine, Stanford, United States; [2]Center for Sleep Sciences and Medicine, Stanford University School of Medicine, Stanford, United States; [3]Department of Molecular and Cellular Physiology, Stanford University School of Medicine, Stanford, United States; [4]INSERM 1024, Ecole Normale Supérieure, Paris, France; [5]Allen Institute for Brain Science, Seattle, USA

**Abstract** The distribution of proteins within sub-synaptic compartments is an essential aspect of their neurological function. Current methodologies, such as electron microscopy (EM) and super-resolution imaging techniques, can provide the precise localization of proteins, but are often limited to a small number of one-time observations with narrow spatial and molecular coverage. The diversity of synaptic proteins and synapse types demands synapse analysis on a scale that is prohibitive with current methods. Here, we demonstrate SubSynMAP, a fast, multiplexed sub-synaptic protein analysis method using wide-field data from deconvolution array tomography (ATD). SubSynMAP generates probability distributions for that reveal the functional range of proteins within the averaged synapse of a particular class. This enables the differentiation of closely juxtaposed proteins. Using this method, we analyzed 15 synaptic proteins in normal and Fragile X mental retardation syndrome (FXS) model mouse cortex, and revealed disease-specific modifications of sub-synaptic protein distributions across synapse classes and cortical layers.

*For correspondence: drwonder@stanford.edu

**Competing interests:** The authors declare that no competing interests exist.

## Introduction

Synapses are central to neural circuit function and dysfunction, but their compositional diversity (there are hundreds of distinct protein species) and structural compactness (they average a single micron in size) have prevented their complete molecular analysis (*Emes and Grant, 2012*; *Heuser and Reese, 1977*; *O'Rourke et al., 2012*; *Chen et al., 2008*; *Sheng et al., 2012*). Many techniques have been developed to visualize synaptic proteins through the nanometer localization of single proteins. These include structured illumination microscopy (SIM; which can locate sub-cellular structures to ~100 nm resolution) (*Gustafsson et al., 2008*; *Gustafsson, 2000*), fluorescence localization microscopy (PALM/STORM; which can examine sub-cellular to single molecules at 10–50 nm resolution) (*Watanabe et al., 2011*; *Dani et al., 2010*), stimulated emission depletion microscopy (STED; which can be used to visualize sub-cellular to sub-synaptic structures at 10–50 nm resolution) (*Wijetunge et al., 2014*; *Willig et al., 2006*, *2007*) and electron microscopy (EM; which can be used to locate single molecules at 1–5 nm resolution) (*Dani et al., 2010*; *Specht et al., 2013*; *Micheva and Beaulieu, 1995*; *Kubota et al., 2007*). These methods can resolve molecules with exquisite precision, but they are time consuming and struggle with both spatial coverage and molecular depth. This current work introduces a new computational method (SubSynMAP: SUB-SYNaptic, Multiplexed Analysis of Proteins) that quantifies the nanometer distribution of multiplex proteins across large regions of the brain with synapse population specificity.

SubSynMAP is a novel combination of previously developed volumetric super-resolution imaging of deconvolution array tomography (ATD) (*Wang and Smith, 2012*; *Micheva and Smith, 2007*), EM ratified computation synapse classification (*Collman et al., 2015*; *Wang et al., 2014*), and a new synapse alignment procedure. It builds protein localization profiles of the average synapse that reveals the multiplexed, functional distributions of dozens of proteins. ATD uses a combination of physical sectioning and deconvolved wide-field imaging to generate an unparalleled combination of acquisition speed, volumetric resolution (100 nm by 100 nm by 50–70 nm) and molecular multiplexing (*Wang and Smith, 2012*; *Micheva and Smith, 2007*). Using ATD, different populations of synapses can be classified from large regions of the cortex based on molecular markers (*Wang et al., 2014*; *Micheva et al., 2010*; *Allen et al., 2012*; *Hiu et al., 2016*). These EM verified algorithms can be used to extract different classes of synapses (excitatory, inhibitory and their subclasses) at large numbers (>1 million) with low error rates (false positive 5%, false negative 12% – for more on antibody verification and error rates see 'Methods') (*Wang et al., 2014*; *Allen et al., 2012*; *Hiu et al., 2016*). Moreover, these algorithms return the molecular features of each synapse organized around a spatial vector that defines the presynaptic bouton to post-synaptic density (*Wang et al., 2014*; *Allen et al., 2012*; *Hiu et al., 2016*) (see *Figure 1E*).

This synaptic-vector is made possible because proteins within presynaptic and post-synaptic compartments are spatially localized in relation to their function (*Heuser and Reese, 1977*; *Chen et al., 2008*; *Dani et al., 2010*; *De Camilli, 1983*; *Hunt et al., 1996*). While individual synapses can vary greatly in their structure and molecular composition, on average, the synapse remains an ordered structure with a specific distribution of protein populations (*Heuser and Reese, 1977*; *Chen et al., 2008*; *Dani et al., 2010*; *De Camilli, 1983*; *Hunt et al., 1996*). These protein distributions are dictated by their functions (*Park et al., 2012*). Thus, the presynaptic proteins required for transmitter release, such as bassoon (BSN), are on average located closer to the active zone; andproteins that are required to load transmitter into vesicles, such as vesicular glutamate transporter 1 (VGluT1), are distributed broadly within the presynaptic vesicle pool (*Dani et al., 2010*; *Micheva et al., 2010*; *El Mestikawy et al., 2011*). In the post-synapse, transmitter receptors, such as ionotropic glutamate receptor 2 (GluR2), are localized to the synaptic cleft, and post-synaptic regulatory molecules, such as fragile x mental retardation protein (FMRP), are situated further away from the cleft (*Chen et al., 2008*; *Dani et al., 2010*; *Dosemeci et al., 2007*; *Bear et al., 2004*).

To make possible the analysis of the spatially multiplexed proteome of the synapse, we demonstrate SubSynMAP, a method that aligns and integrates tens of thousands of individual synapses within a brain region. This method extracts the averaged protein distributions within molecularly defined synapses drawn from large populations, and allows for the sub-synaptic visualization of multiplexed proteins. For example, in accordance to previous EM and STORM measurements (*Dani et al., 2010*; *Mineur et al., 2002*), BSN and VGlut1 distributions pre-synaptically peaked 25 and 60 nm, respectively, from the cleft, whereas GluR2 peaked 20 nm post-synaptically from the cleft. These spatial distributions are also useful for the characterization of novel proteins within specific synapse populations. We found that FMRP is distributed both pre- and post-synaptically, while its autosomal homologs FXR1 and FXR2 are predominantly distributed post-synaptically. Interestingly, in fragile X mental retardation 1 knock out mice (*Fmr1* KO), a well-accepted model of FXS (*Bear et al., 2004*; *Bassell and Warren, 2008*; *Graziano et al., 2008*) in whichFMRP is no longer expressed, we found that synaptic FXR2 distribution is disrupted while FXR1 remains unperturbed. This suggests that proper synaptic FXR2 localization requires FMRP and a FXR2 deficit could be part of the pathophysiology of FXS synapse dysfunction.

## Results

### Computational alignment of synapse populations reveals the probability distribution of synaptic proteins

ATD provides exquisite volumetric resolution of spatially registered synaptic proteins (*Wang and Smith, 2012*). These punctate protein data allow for the recognition and classification of synapses based on the colocalization and spatial orientation of known pre- and post-synaptic proteins (*Figure 1* and *Figure 1—figure supplement 1*) (*Wang et al., 2014*). Specific synaptic proteins can be used to define synaptic axes with a defined orientation (post-synaptic to presynaptic) by connecting

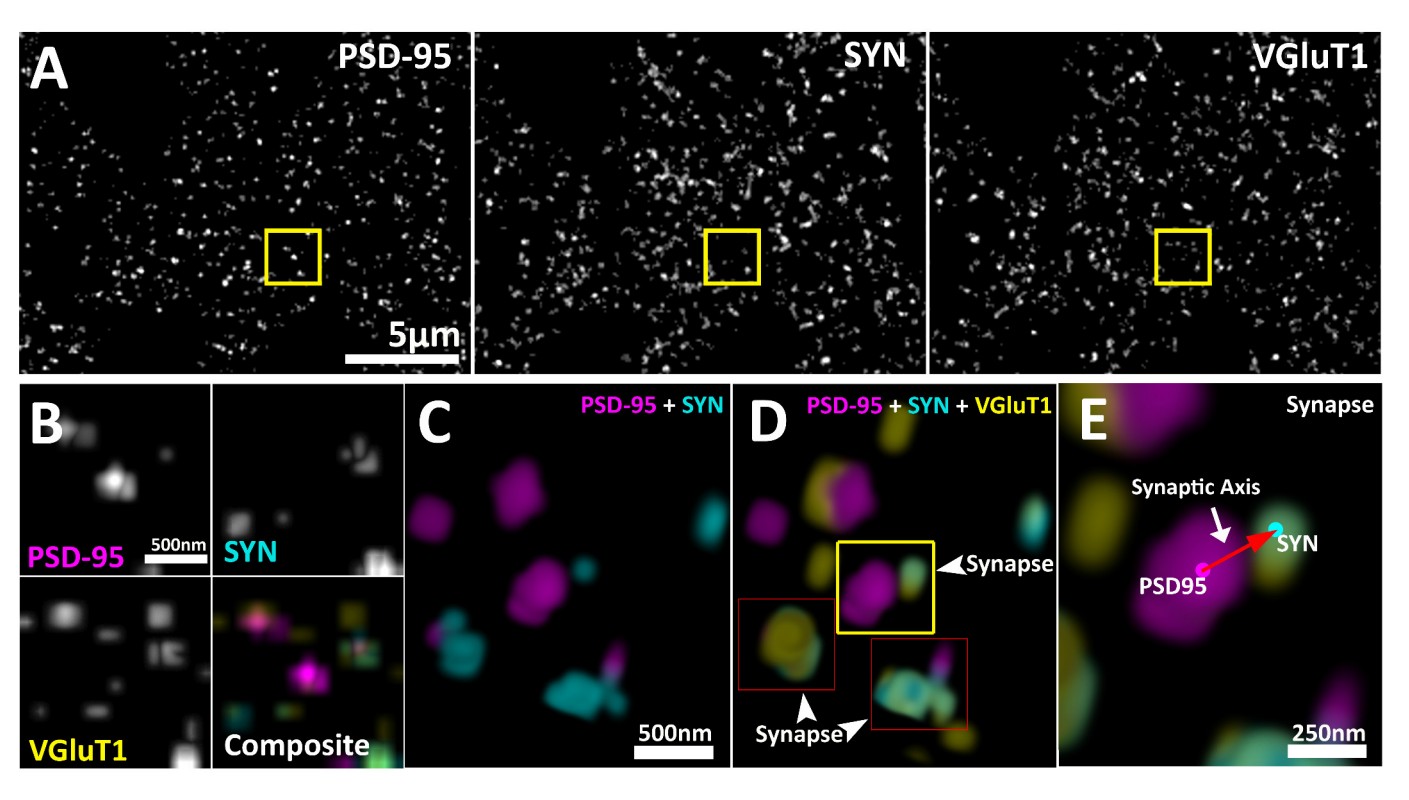

**Figure 1.** Multiplexed visualization of synaptic proteins in AT. (**A**) Representative maximum (max) projections (25 μm by 20 μm by 1.5 μm) of PSD95, SYN and VGluT1. Scale bar = 5 μm. (**B**) Up-sampled view of the highlighted regions in panel (**A**) that represents a volume of 1.5 μm x 1.5 μm x 1.5 μm. Scale bar = 500 nm. (**C–E**) Composite 3D rendering and rotational perspectives of PSD95 and SYN, and PSD95, SYN and VGluT1 volumes highlighted in panel (**B**). Scale bar = 500 nm. (**D**) Arrow heads point to putative synapses revealed by the structured localization of synaptic proteins. (**E**) Expanded panel of the central synapse in panel (**C**) (Yellow box). Red arrow illustrates a synaptic axis drawn from the center of mass of a PSD95 punctum to a SYN punctum. Scale bar = 250 nm.

The following figure supplement is available for figure 1:

**Figure supplement 1.** Array tomography generates spatially registered volumes of multiplexed protein data.

centers of mass from post-synaptic density protein 95 (PSD95) to presynaptic synapsin-1 (SYN) for glutamatergic synapses (*Figure 1E*), and post-synaptic gephyrin (GPHN) to presynaptic vesicular GABA transporter (VGAT) for GABAergic synapses. The centers of mass are calculated from segmented puncta objects as defined by a thresholded, three-dimensional (3D) connected component analysis (see 'Methods'). Centers of mass are used in our calculations to facilitate computation of the over 9 million protein puncta collected across the 13 synaptic proteins in our ATD volumes (*Figure 1—figure supplement 1*; see 'Methods' for further description of the datasets).

These synaptic vectors, which are oriented in many directions in the tissue volumes (*Figure 2A*), are computationally aligned into one coordinate system, creating a registered synaptic volume (*Figure 2B and C*, see 'Methods'). This volume contains the aligned centers of all protein puncta located within a 1 μm radius of the origin (PSD95 weighted center) of each synaptic vector (*Figure 2C and D*). The raw 3D histograms of these protein volumes reveal distinct pre- and post-synaptic orientation (*Figure 3—figure supplement 1* and *Video 1*). To facilitate the interpretation and analysis of these volumes, two-dimensional (2D) histograms are created by rotationally projecting all the centers of a single protein into a 2D plane. The bins of these 2D histograms are half slices of a spherical shell (*Figure 3A*). Rotational projection is used to insure that the distance of all centers from the origin is maintained in the histogram (*Figure 3A*). Moreover, these histograms are generated as a half distribution because there is no rotational constraint along the synaptic axis

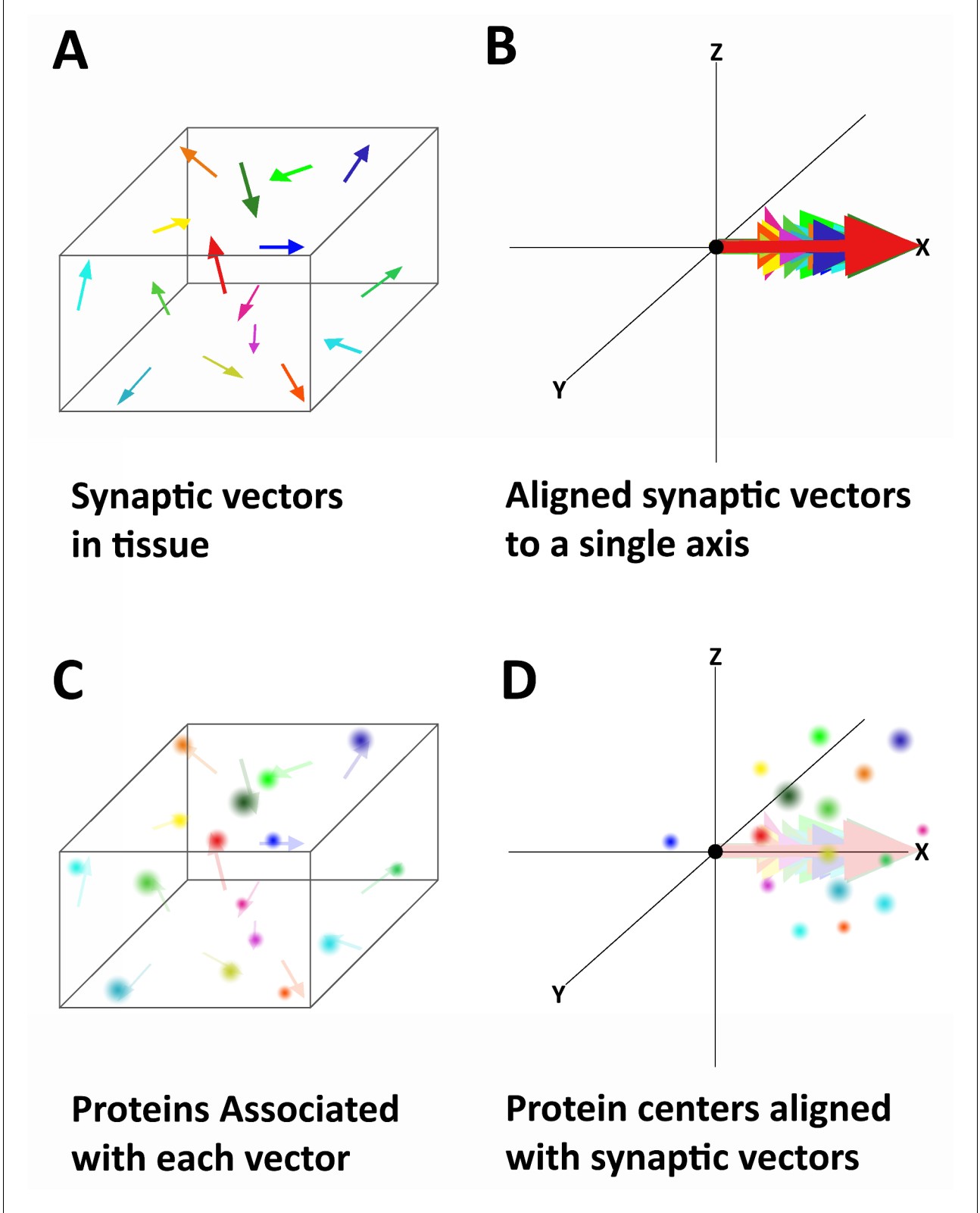

**Figure 2.** Alignment of synaptic axes into a single spatial coordinate system. (**A**) Diagrammatic representation of the random orientation of synaptic axes in normal cortical tissue. (**B**) All the axes are translated to a single origin and then computationally rotated on to a single axis. This creates a single space in which all of the synapses are aligned along the synaptic axis. (**C**) Proteins associated with each synapse are disorganized in space, but have a

*Figure 2 continued on next page*

*Figure 2 continued*

structured relationship with the synaptic axis. This is difficult to see in tissue because the synapses are not ordered. (**D**) Once the synapses are aligned, the proteins associated with each synapse are also aligned and any non-random structure of their relationship to the synaptic axis is revealed.

(*Figure 3B*). Thus, to better conceptually the full synaptic distribution of the protein, this half distribution is mirrored to create a full distribution (*Figure 3B*).

Due to the rotational nature of the projection into 2D, the volume of each histogram bin increases as a function of its distance from the origin (*Figure 3C and D*). This means that the probability that a protein center falls randomly into any given bin increases as a function of the bin's distance from the origin (*Figure 3E*). Thus, to remove this effect, the histogram is normalized with the bin volumes to create a density histogram in which the random probability that a protein center will land within a bin is equal across all bins (*Figure 3F*, see 'Methods'). The density histogram reveals the probability that a protein center will be found at a position in a given synapse of that population. Thus, this distribution is representative of the functional spatial-range of a synaptic protein and defines quantitatively the potential transit area of that protein within the average synapse of a given class.

As an example, in *Figure 3*, vesicular glutamate transporter 1 (VGluT1) protein centers from nearly 40,000 classified VGluT1 class synapses of layer 4 mouse somatosensory cortex (for classification, see 'Methods') are binned into a density histogram. VGluT1 is one of two presynaptic glutamate transporters that are essential to the loading of glutamate into cortical synaptic vesicles, the other being VGluT2 (*Fremeau et al., 2001*; *Kaneko and Fujiyama, 2002*; *Siksou et al., 2007*). The presynaptic localization of VGluT1 is reflected by the histogram (*Figure 3E and F*). Moreover, the histogram is also able to resolve the exclusion of VGluT1 from the post-synaptic zone, which is demonstrated by the below background dark area in the post-synaptic region (*Figure 3E and F*). To further confirm the fidelity of the method in characterizing synaptic protein distributions, the association of the VGluT1 centers and their synaptic axes were randomized. The resulting random raw histogram appears similar to the bin volume histogram (*Figure 3D and G*), as the probability that a center will falls within a bin in an unbiased dataset is solely dependent upon the volume of the bin, thus the resulting normalized density histogram is flat and featureless (*Figure 3H*).

## Empirical protein probability distributions reveal nanometer localization differences of synaptic proteins

Density histograms, such as the one in *Figure 3F*, can be converted to a probability histogram by dividing the number of protein centers per bin by the number of synapses. This generates an estimated percent probability that a protein center can be found at a position in space in the average synapse of that class; for example, a probability that VGluT1 proteins will be found in VGluT1 class synapses in layer 4 (*Figure 4A*). The shape of the VGluT1 empirical probability distribution in Layer 4 VGluT1 synapses is asymmetrical in regards to the synaptic axis (PSD95 to SYN), with a wider distribution along the axis and a narrower distribution perpendicular to it (*Figure 4A*). This is likely to reflect a distribution of synaptic vesicles in which the major pool of vesicles, in excitatory synapses, resides axially behind the active zone (*Heuser and Reese, 1977*; *Park et al., 2012*; *Mineur et al., 2002*). To further simplify the visualization and statistical

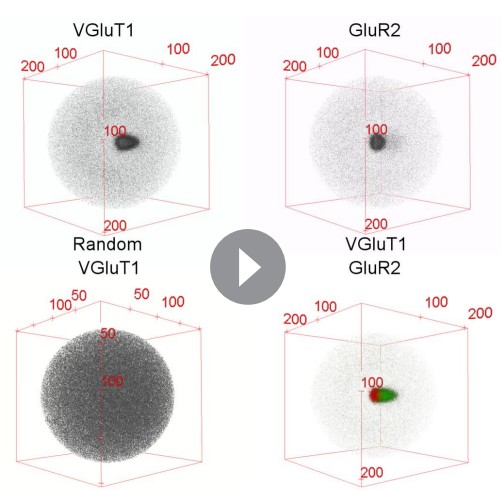

**Video 1.** Histograms of SubSynMAP data. 3D visualization of VGluT1, GluR2, VGluT1 and GluR2 composite and of randomized VGluT1 data. The bins of the 3D histograms are of equal volume, 10 nm$^3$. Note that the random data produces a sphere and the composite VGluT1 and GluR2 showed pre and post-synaptic alignment.

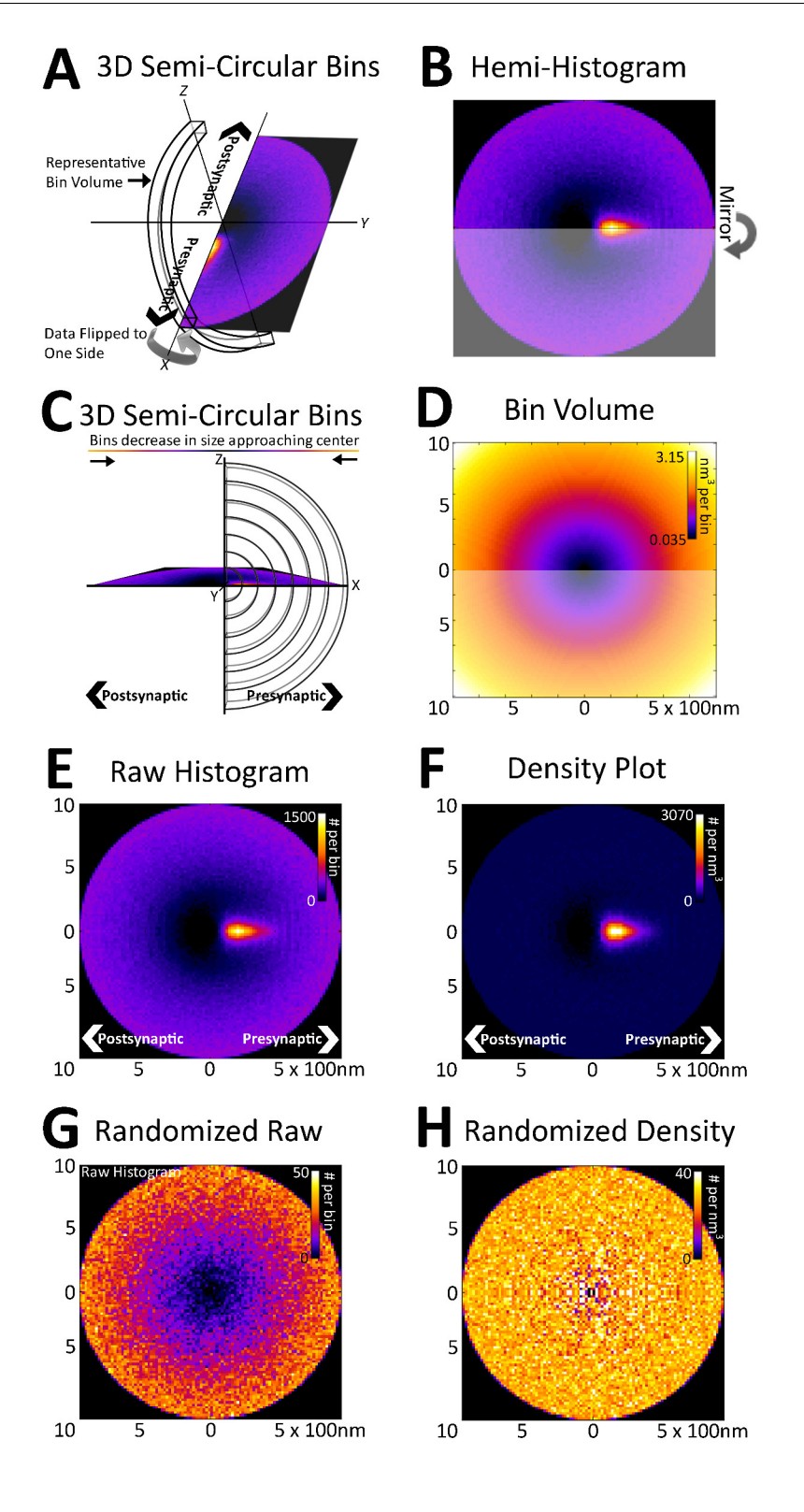

**Figure 3.** Registration of synapses reveals a nanometer-resolution probability distribution of synaptic proteins. (**A**) Protein centers from registered synapses are summed using bins that are slices of spherical shells to maintain the distance of each center to the origin. As there is no rotational invariance along the x-axis, the data are binned into a hemi-histogram. (**B**) This hemi-histogram is then mirrored to provide a better representation of the full synapse. (**C**) A profile view of the bins further depicts the size change of the bins as they move from the center. (**D**) A graphical depiction of bin

*Figure 3 continued on next page*

*Figure 3 continued*
volumes moving from the center of the figure. (E) This raw 2D histogram is generated for all the VGluT1 centers within a 1 μm radius sphere around the central PSD95 (origin) of VGluT1 class synaptic axes (n = 36,997). (F) The raw histogram is normalized by the calculated volume of each bin to create the density histogram. The density histogram represents the probability of a VGluT1 center landing in that location in the average synapse of that population. (G) The relationship of VGluT1 centers to their associated synaptic axes is randomized and the resulting histogram is plotted, which looks very much like (D). (H) The density histogram created from these randomized data shows no probability bias, thus creating a flat and featureless image.
The following figure supplement is available for figure 3:

**Figure supplement 1.** 3D histograms of SubSynMAP data.

comparison of these distributions, the 2Dprobability histograms are rotationally projected to a one-dimensional (1D) plot (*Figure 4A*). In this plot, the origin is the PSD95 centers of the nearly 40,000 VGluT1 synapses registered in the dataset. The putative synaptic cleft (red dashed line, *Figure 4*) is 50 nm from the origin, as estimated from previous EM and super-resolution measurements (*Dani et al., 2010*; *Mineur et al., 2002*). Assessing the plot, the VGluT1 distribution dips in the post-synaptic zone, demonstrating the expected anti-correlation of VGluT1 localization to the post-synapse. Furthermore, the distribution peaks at ~125–200 nm from PSD95 or about 75–150 nm from the cleft (*Figure 4A*). These measurements corroborate previous analysis of presynaptic vesicle pools using EM (*Mineur et al., 2002*) or super-resolution microscopy, demonstrating that this method provides an accurate visualization of protein distribution in synapses.

Using this method, we demonstrate that glutamate receptors of both the AMPA and NMDA sub-type (GluR2 and NR1), which reside in the post-synaptic active-zone membrane, have a distribution that peaks near the synaptic cleft, where they are expected to reside (*Figure 4B,D,F*). More importantly, while the empirical glutamate receptor distributions are similar to each other, they are significantly different from the empirical presynaptic protein distributions (*Figure 4A–F*, Chi-square test $p < 0.01$, see 'Methods'). Moreover, while the empirical distribution of presynaptic vesicular proteins VGluT1 and synaptophysin (SYNPH) are similar (*Figure 4A and C*,$p > 0.05$); they are significantly different from the empirical distribution of the presynaptic structural protein bassoon (BSN) (*Figure 4A,C,E*, $p < 0.01$). BSN is an important cytoskeletal component of vesicle priming, which was demonstrated by both EM and STORM to be located near the active zone (*Dani et al., 2010*; *Mineur et al., 2002*). Thus, our method can reveal with statistical significance the fine-scale distributional differences of synaptic proteins, and can correlate those variations with their function, in for example the vesicle pool or the active zone.

Peri-synaptic astrocytes also play an important regulatory role in synaptic function. This technique is able to visualize astrocytic proteins, which typically form 'halos' around the pre- and post-synapse (*Figure 4G–J*), exactly where peri-synaptic astrocytic processes reside (*Amara and Fontana, 2002*). Interestingly, the astrocytic proteins glial glutamate transporter 1 (GLT1) and glutamine synthetase (GS) both show dips in the synaptic cleft region (*Figure 4G and I*), whereas glutamate aspartate transporter (GLAST) presents a peak at the cleft (*Figure 4H*). This suggests that GLAST, in comparison to GLT1, is more localized to the cleft. This result points to a currently unknown functional separation of the two major cortical, astrocytic-glutamate transporters.

## Multiplexed analysis of empirical synapse protein distribution across synapse classes and cortical layers demonstrates the stability of cortical synapses across layers and classes

This fine-scale analysis, coupled with the spatial and molecular coverage of this technique, allows us the opportunity to quantify whether shared, synaptic proteins are distributed similarly between synapse classes and cortical layers. In *Figure 5*, the distribution of L4 proteins in nearly 100,000 synapses were analyzed across two glutamatergic synapse classes (VGluT1 and VGluT2) and one inhibitory synapse class. VGluT1 represents the majority of cortical–cortical synapses and VGluT2 defines the majority of thalamocortical synapses in the cortex (*Fremeau et al., 2001*; *Kaneko and Fujiyama, 2002*; *Herzog et al., 2001*). It is satisfying to see in our datasets that in both Layers 4 and 5, VGluT1 synapses have little VGluT2 protein and vice versa (*Figure 5A and B* and *Figure 5—figure supplement 1A and B*). This demonstrates the fidelity of the synapse classification and the accuracy of the

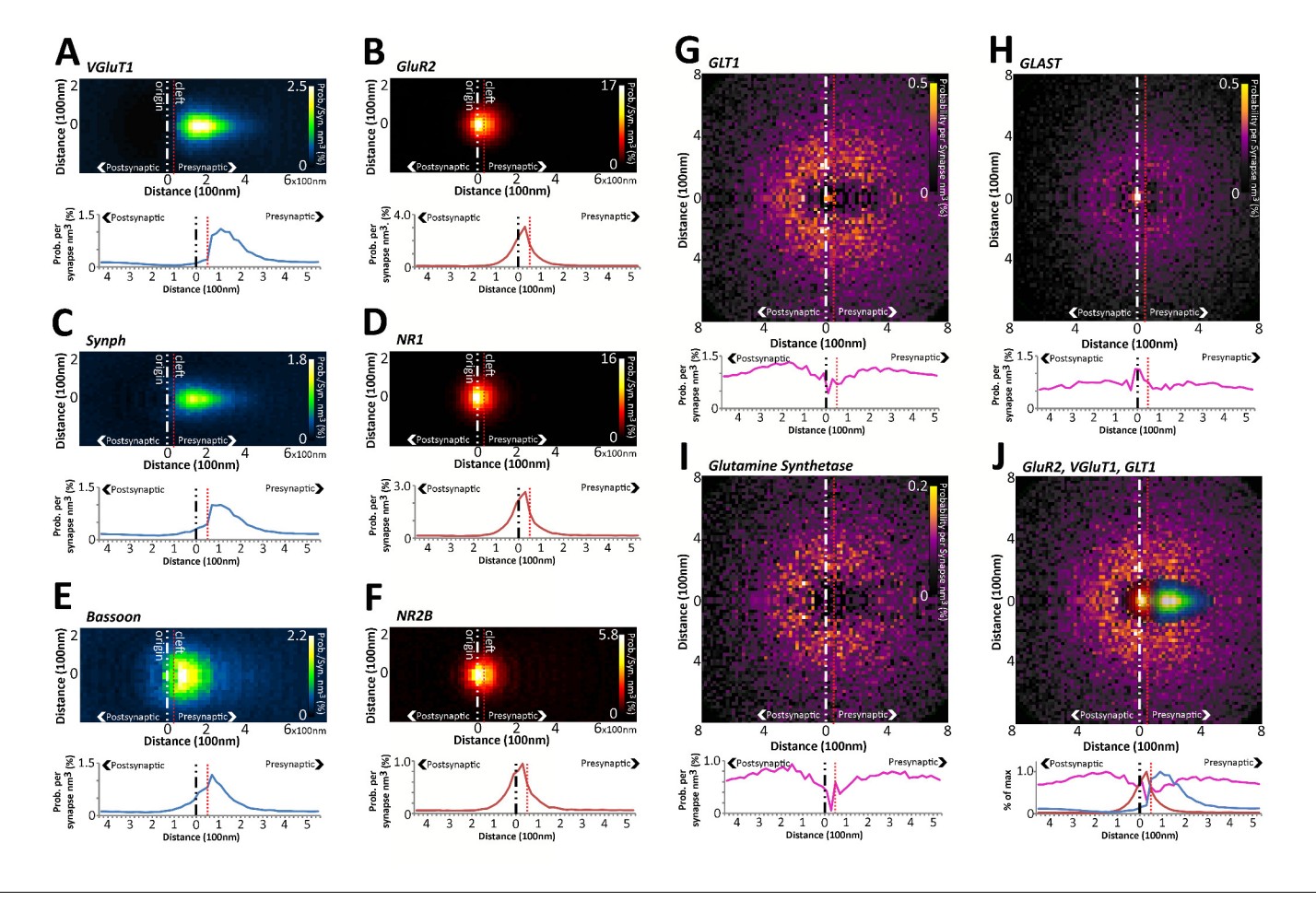

**Figure 4.** Quantitative differences in the nanometer distribution of synaptic proteins in layer 4 VGluT1 synapses. Volume-normalized 2D probability plots were generated from 36,997 classified VGluT1 synapses in Layer 4 of mouse cortex. Each channel reveals specific information regarding the distribution of those proteins in relation to the synaptic axis. The 2D distributions are further reduced to a 1D distribution to ease the visual and statistical comparison of the data. The dashed red line represents the putative location of the synaptic cleft. The y-axis scale is in percentage points, because density is converted to percent probability by dividing the number of protein centers per bin by the number of synapses. This generates an estimated percent probability that a protein center can be found at a position in space in the average synapse of that class. (A–F) Pre- and post-synaptic protein distributions are clearly differentiated. The distributions of presynaptic vesicular proteins (VGluT1 and SYNPH) are significantly different from those of the presynaptic active zone protein BSN. (G–J) Astrocytic proteins are distributed as halos surrounding the synapse center. (J) The exclusion at the center of the GLT1 halo fits the distribution of the presynaptic and postsynaptic proteins nicely.

protein visualization. A point that is further validated by the near-random distribution of VGluT1 and VGluT2 proteins in inhibitory synapses (*Figure 5C* and *Figure 5—figure supplement 1C*), which illustrates that even for a densely expressed protein, such as VGluT1 (*Figure 1*), the method is robust and accurate enough to reveal the lack of association between that protein and the inhibitory synaptic axes.

Upon quantifying the distributional differences of proteins across Layers 4 and 5 of these three synapse classes, there are no significant differences between the two layers when comparing the same protein within the same synapse class (*Figure 5* and *Figure 5—figure supplement 1*, p > 0.01). This is also true of the cross-class comparison between VGluT1 and VGluT2 synapses within the same cortical layer (*Figure 5A and B* and *Figure 5—figure supplement 1A B*, p > 0.01). This demonstrates that the overall synapse architecture of excitatory synapses remains stable between classes and layers, and that inhibitory synapse architecture is stable between layers as well.

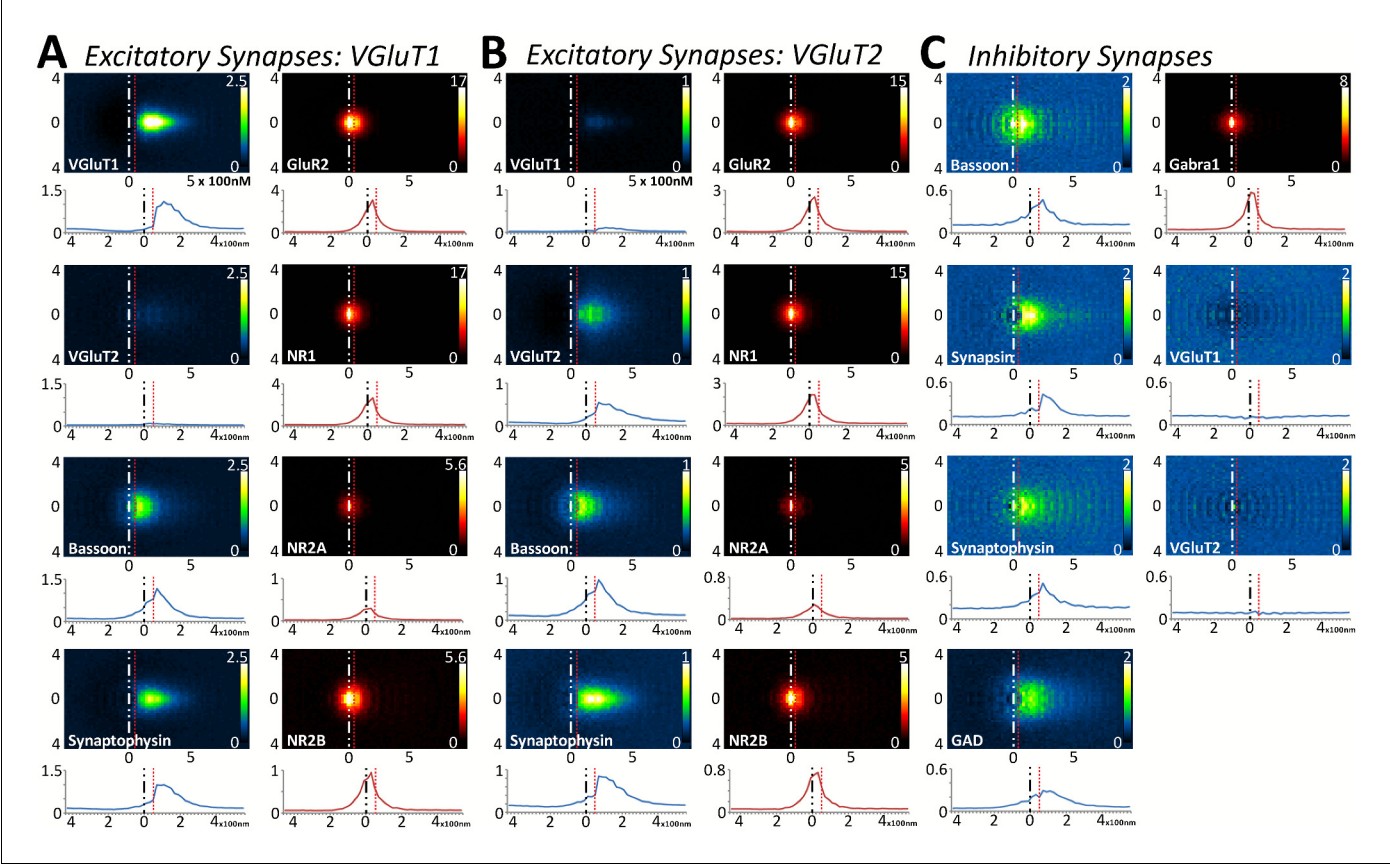

**Figure 5.** Multiplexed analysis of synaptic protein distribution across Layer 4 synapse classes. (**A–B**) Probability distribution of proteins in glutamatergic synapses. (**A**) Probability distributions of proteins surrounding VGluT1 (n = 36,997). (**B**) Probability distributions of proteins surrounding VGluT2 (n = 6047). (**C**) Probability distribution of proteins in GABAergic synapses (n = 8100). Note that VGluT1 synapses have low levels of VGluT2, and VGluT2 synapses have low levels of VGluT1. Inhibitory synapses do not have VGluT1 or VGluT2. In fact, the dip in the center of histogram reveals that VGluT1 is anti-correlated with inhibitory synapses. Source data for the histograms in this figure can be found in *Figure 5—source data 1*.

The following source data and figure supplements are available for figure 5:

**Source data 1.** Multiplexed analysis of synaptic protein distribution across Layer 4 synapse classes.

**Figure supplement 1.** Large-scale analysis of synaptic protein distribution across Layer 5 synapse classes.

**Figure 5–figure supplement 1–source data 1.** Multiplexed analysis of synaptic protein distribution across Layer 5 synapse classes.

Interestingly, this population-level analysis of excitatory and inhibitory synapses also uncovers the stability of some common elements: BSN distribution is not different between inhibitory and excitatory synapses (*Figure 5*, *Figure 5—figure supplement 1*, p > 0.01). However, we found that this is not true for all proteins shared by both synapse classes. For example, SYNPH distribution is significantly different depending upon whether it is within an inhibitory or an excitatory presynaptic terminal (*Figure 5*, *Figure 5—figure supplement 1, p* < 0.01). This suggests that the structure of the presynaptic release architecture remains similar across the two synapse types (as demonstrated by BSN), but the structure of the synaptic vesicle pools differs between inhibitory and excitatory synapses (as demonstrated by SYNPH and consistent with previous findings in both EM (*Khanmohammadi et al., 2015*) and electrophysiology [*Moulder et al., 2007*]).

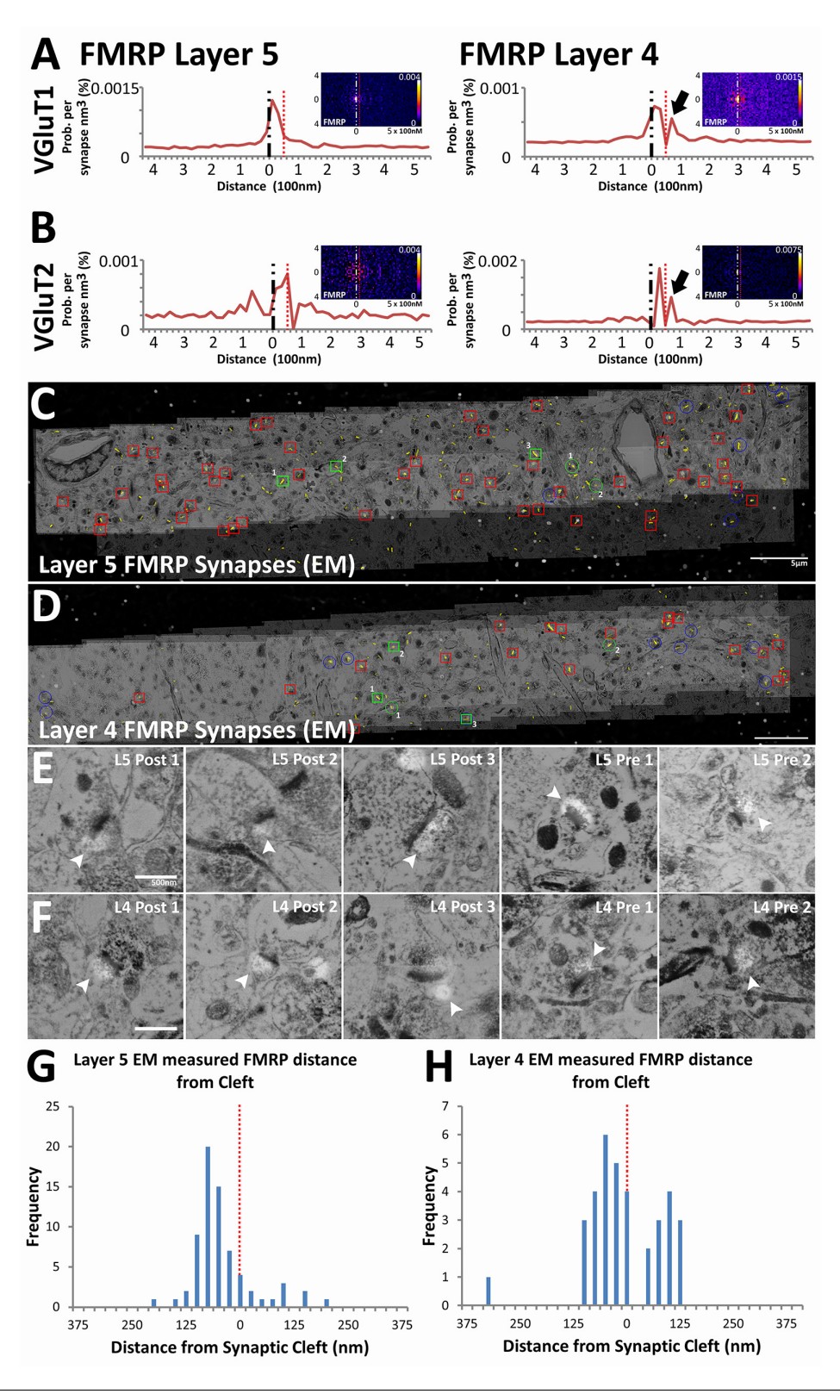

**Figure 6.** Presynaptic localization of FMRP is synapse class dependent. (**A**) Synaptic distribution of FMRP in VGluT1 synapses of Layers 5 (n = 6877) and 4 (n = 11,346). FMRP peaks in the post-synapse for both Layer 5 and 4 VGluT1 class synapses, but only Layer 4 has a presynaptic peak too (black arrow). (**B**) Synaptic distribution of FMRP in VGluT2 synapses of Layers 5 (n = 487) and 4 (n = 1797). FMRP also peaks in the post-synapse in both Layer 5 and Layer 4 VGluT2 synapses. (**C, D**) Single-section images of EM micrographs aligned with single-section images of fluorescent FMRP staining. Red boxes

*Figure 6 continued on next page*

*Figure 6 continued*

are hand classified synapses with an associated FMRP punctum. Green boxes are exemplar synapses that are shown below in (E) and (F). (E, F) Example EM synapses with fluorescent FMRP labeling. Note the synaptic vesicles and PSD that allows for the delineation of the pre- and post-synapse. All images are single sections for both EM and fluorescence images. (G, H) Histograms of FMRP distance from the synaptic cleft (at 0), the presynaptic side is on the right and the postsynaptic side is on the left. (G) FMRP in Layer 5 is highly post-synaptic. (H) FMRP in Layer 4 is more evenly split between pre- and postsynapses. The EM histograms are also narrower, because of the z direction clipping of the EM dataset (210 nm in this figure), as compared to the light level dataset (2.8 µm in previous figures). Source data for the histograms in this figure can be found in *Figure 6—source data 1* and *2*.

The following source data is available for figure 6:

**Source data 1.** Presynaptic localization of FMRP is synapse-class dependent.
**Source data 2.** Presynaptic localization of FMRP is synapse class dependent.

## Layer-specific sub-synaptic localization of FMRP

We validated the use of our method on novel synaptic proteins by generating insight into the physiology of synapses in fragile X syndrome (FXS). FXS is a leading cause of congenital mental retardation that is mediated by the loss of FMRP, an RNA-binding protein that inhibits activity dependent translation in dendrites (*Darnell et al., 2011*; *Huber et al., 2002*). Canonically, FMRP is considered to be a post-synaptic protein, but recent reports also suggest that it can be detected in presynaptic terminals (*Christie et al., 2009*; *Akins et al., 2012*). In our analysis, FMRP is indeed localized post-synaptically in both VGluT1 and VGluT2 synapse classes across Layers 4 and 5 (*Figure 6A and B*). Moreover, the empirical distribution of post-synaptic FMRP, unlike that of the post-synaptic glutamate receptors, peak away from the synaptic cleft (*Figure 6A and B*), which concurs with its role as a post-synaptic regulatory molecule (*Darnell et al., 2011*; *Huber et al., 2002*). Interestingly, in Layer 4 synapses only, a second FMRP peak can be detected presynaptically (*Figure 6A and B*, black arrows). This peak is not pronounced in the empirical distribution of Layer 5 synapses (*Figure 6A and B*), which suggests that although post-synaptic FMRP localization is a general feature, the bulk of presynaptic FMRP is directed to Layer 4 rather than to Layer 5.

Validation of this result was confirmed using EM conjugate AT (*Collman et al., 2015*). Four datasets were generated (for Layers 4 and 5 of both WT and *Fmr1* KO mice), each covering a volume of 60 µm x 10 µm x 210 nm and consisting of both EM and light level immunohistochemistry (VGluT1, FMRP and FXR2). Using hand classification, we identified 115 WT Layer 4 synapses, 199 WT Layer 5 synapses, 113 KO Layer 4 synapses and 118 KO Layer 5 synapses. After the alignment of EM and light level data (*Figure 6C and D*), pre- and post-synapses are identified in EM using the presence of synaptic vesicles and the PSD (*Figure 6E F*). Using conjugate FMRP light level data in combination with hand-classified synapses, FMRP-containing synapses were identified (*Figure 6E and F*). The distance to cleft from the FMRP punctum center was measured and graphed on a histogram (*Figure 6G and H*). The histogram of FMRP distance to cleft in EM (*Figure 6G and H*) clearly reproduces the wide-field, synapse alignment result (*Figure 6A*). The EM histograms are narrower in distance because the EM volumes in the z-axis are significantly smaller than an actual synapse (210 nm), whereas this is not the case in the wide-field dataset (EM volumes are 2.8 µm). In our EM data, Layer 4 clearly has more presynaptic FMRP (230% more) than layer 5, which corroborates our SubSynMAP analysis.

## FMRP-dependent localization of FXR2P

Fragile X related proteins 1 and 2 (FXR1P and FXR2P) are nearly structurally identical autosomal homologs of FMRP (*Tamanini, 1997*). Yet, their involvement in FXS and their synaptic localization and function is mostly unknown. The application of our synapse analysis method demonstrates that, across synapse classes and cortical layers, FXR1P and FXR2P unlike FMRP are preferentially localized to the post-synaptic region (*Figure 7A–D*, *Figure 7—figure supplement 1A–D*); but like FMRP, the distributions of FXR1P and FXR2P peak away from the synaptic cleft (*Figure 7A–D* and *Figure 7—figure supplement 1A–D*). Furthermore, by assessing the distributions of these proteins in *Fmr1* KO animals, we reveal that the loss of FMRP does not affect the synaptic localization of FXR1P (*Figure 7—figure supplement 1, p* > 0.01), but causes a significant depletion of post-synaptic FXR2P in

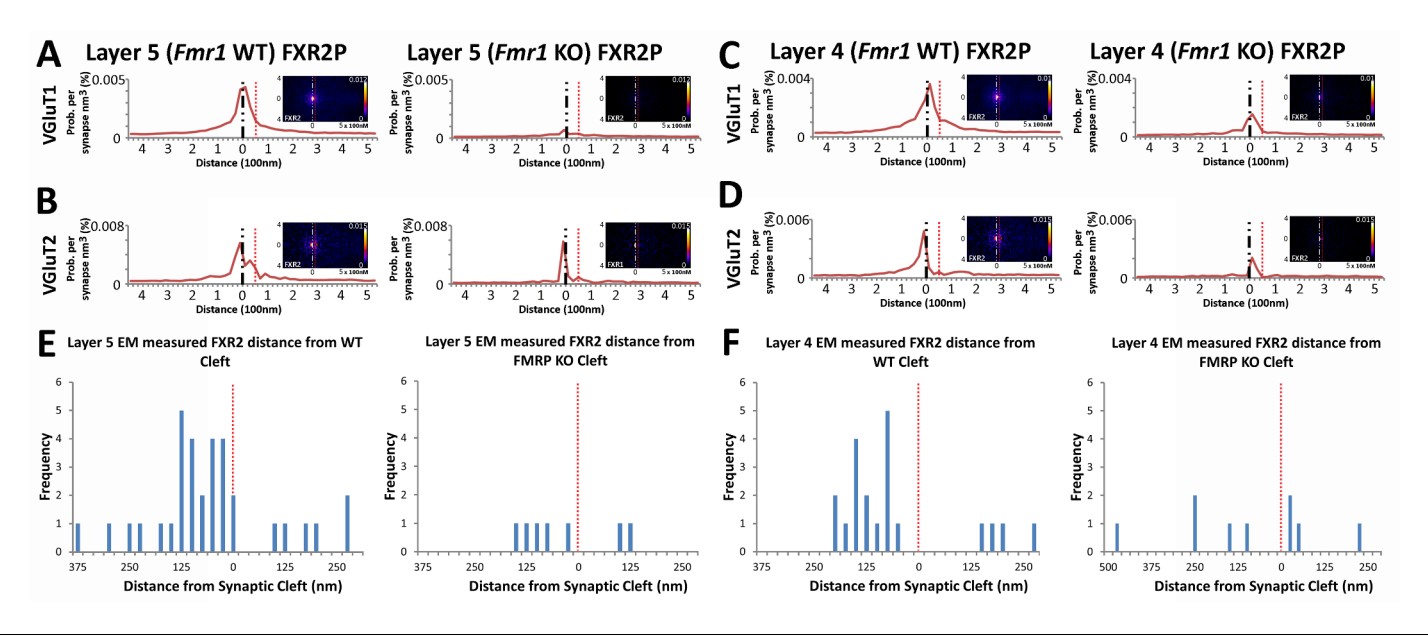

**Figure 7.** FXR2P synaptic localization is FMRP dependent. (**A**) Synapse distribution of FXR2P in Layer 5 VGluT1 class synapses in both KO (n = 5381) and WT (n = 9481). FXR2P post-synaptic localization is dramatically reduced in Layer 5. (**B**) Synapse distribution of FXR2P in Layer 5 VGluT2 synapses in both KO (n = 563) and WT (n = 1483). FXR2P post-synaptic localization is reduced. (**C**) Synapse distribution of FXR2P in Layer 4 VGluT1 synapses in both KO (n = 5341) and WT (n = 9917). (**D**) Synapse distribution of FXR2P in Layer 4 VGluT2 synapses in both KO (n = 1207) and WT (n = 1783). Significance of distribution difference is tested via chi-squared two-sample tests (see 'Methods', p < 0.01). (**E, F**) EM distance histograms of FXR2 from synaptic cleft (at 0). Both Layers 4 and 5 show dramatic reduction of synaptic FXR2P. Source data for the histograms in this figure can be found in *Figure 7—source data 1* and *2*.

The following source data and figure supplements are available for figure 7:

**Source data 1.** FXR2P synaptic localization is FMRP dependent.

**Source data 2.** FXR2P synaptic localization is FMRP dependent.

**Figure supplement 1.** FXR1P synaptic localization is not FMRP dependent.

**Figure supplement 2.** Conjugate EM visualization and localization of FXR2P synapses.

both Layers 4 and 5, as measured both in SubSynMAP (*Figure 7A–D*, p < 0.01) and EM (*Figure 7E and F*, *Figure 7—figure supplement 2*). This suggests that the synaptic localization of FXR2P depends on FMRP. However, we do not know whether this coupling of expression proceeds through the FXR2P–FMRP heteromeric complexes (*Christie et al., 2009*; *Tamanini et al., 1999*; *Ceman et al., 1999*) or through FMRP regulation of FXR2P mRNA (*Ascano et al., 2012*). Importantly, it suggests that FXR2P localization could contribute to the synaptic pathology seen in FXS.

## Discussion

We demonstrated a highly multiplexed method (SubSynMAP) for the visualization and quantification of the nanometer-resolution distribution of proteins within synapse populations. SubSynMAP uses the statistical power of large datasets to reveal discrete variations in the sub-synaptic distribution of synaptic proteins. For example, we can differentiate presynaptic vesicular proteins (such as SYNPH, VGluT1) from presynaptic active zone proteins (such as BSN). The comparison of these distributions can provide meaningful insight into the potential function of these proteins. For example, the greater localization to the synaptic cleft region of GLAST as compared to GLT1 suggests that GLAST plays a more important role in the direct clearing of glutamate in the cleft, whereas GLT1 localization

reflects its known function as a peri-synaptic regulator of metabotropic glutamate receptor signaling (*Bar-Peled et al., 1997*; *Tzingounis and Wadiche, 2007*; *Reichelt and Knöpfel, 2002*). Furthermore, our method is dramatically faster and more cost effective than traditional sub-synaptic protein analysis. Our work using EM to optimize, image and analyze 545 synapses for two proteins (FMRP and FXR2P) took over a year and tens of thousands of dollars , whereas the optimization, imaging and analysis of over 500,000 synapses for 15 proteins, took less than 1 month using SubSynMAP.

SubSynMAP is easier and faster than traditional super-resolution methods, such as EM, but its goal is to complement those methods, not to replace them. Methods such as EM provide detailed visualization of sub-synaptic proteins with exceptional resolution (<5 nm), SubSynMAP cannot be measured along those traditional resolution metrics, nor can its data be interpreted along those lines. In SubSynMAP, the resolution does not come from a single high-resolution snap shot, but rather from the combined distribution of tens of thousands of observations at lower resolution (100 $nm^3$). The distributions in SubSynMAP are not meant to represent the location of any one protein at any one time, which is information that EM can provide. Instead, it represents the likelihood of finding that protein in that location of the prototypical synapse of a specific class if you make many observations. Thus, it does not resolve a protein's location per se, as one would in immunoEM, which can be used to resolve a protein on synaptic vesicles. SubSynMAP can, however, tell you, by comparing the distribution of a particular protein with that of known vesicle proteins, whether a protein is likely to be a vesicle protein.

Another feature that warrants thought is the selection of the protein markers that are used to generate the synaptic axis. In this paper, all excitatory synapses use an axis described by a line drawn from the center of a PSD95 puncta to a synapsin puncta, whereas inhibitory synapses use a line from gephyrin to VGAT. In practice, we found no difference in resultant SubSynMAP histograms from axes generated from PSD95 to vesicle proteins; for example, there are no differences in the histogram for synaptic vesicle protein 2 (SV2) and that for vesicle pool related structural proteins such as synapsin. Nevertheless, one cannot fully rule out the possibility that data generated from different synaptic axes are not identical. Thus, comparison across such different axes should be verified with EM when possible.

This approach to the description of the substructure of a synapse not only provides protein localization data but also observational power. Traditional super-resolution or ultra-structural analysis routinely involves small sample sizes in terms of animals, synapse number and imaged volume. In this study, over a half million synapses from 20 animals across two cortical layers were analyzed, creating an overall image that is more likely to reflect the reality of the synapses studied. The extent of the coverage, both in numbers and in space, allows the analysis of synapse classes that are rare in relation to the total synapse population. For example, Layer 5 VGluT2 synapses only represent about eight percent of the total synapse population, and of that eight percent, only about a quarter have FMRP in them. Thus, a direct sample of 100 synapses in Layer 5 using traditional methods will yield, at most, two VGluT2 synapses with FMRP, a quantity that is mostly anecdotal. In our study, we are able to analyze a total of 487 Layer 5 VGluT2 FMRP synapses, which provides a more robust measure of their structure.

Structural modification of synaptic proteins is hypothesized to be an integral part of most neurological diseases, yet there is little actual structural data supporting this. This data gap is caused by the potential diversity of synapse modifications in neurological diseases, which requires large sample sizes to obtain significant numbers of similarly perturbed synapses. This is extremely challenging for current methods of structural analysis. Our method can fill this data gap. We demonstrated in *Fmr1* KO mice that we are able to reveal specific changes in protein locations mediated by FMRP loss within defined synapse classes. We localized proteins (FXR1P and FXR2P), within normal and diseased contexts, whose synapse localization was previously unknown. Finally, the large-scale quantification of small, complex, stereotypically structured protein complexes is not just a problem in neuroscience, but in biology in general. SubSynMAP, while designed for synapse analysis, can be generalized to a much wider array of cellular structures and should provide a new level of analytical power for other disciplines such as immunology and cancer studies that require subcellular resolution investigations of large tissues.

## Methods

### Preparation of ATD ribbon arrays

Tissue preparation, array creation and immunohistochemistry are described in detail in a previous publication (*Micheva and Smith, 2007*). In short, a small piece of cortical tissue (~2 mm high x 1 mm wide x 1 mm deep) from the somatosensory cortex of the mouse brain is microwave-fixed in 4% paraformaldehyde. The fixed tissue is then dehydrated in graded steps of ethanol, and then embedded in LR White resin overnight at 50°C. The embedded tissue is sectioned on an ultramicrotome at a thickness of 70 nm and placed as a ribbon array directly on gelatin- or carbon-coated glass coverslips. For conjugate EM and array tomography, tissues are embedded in Lowicryle instead of LR White. The Lowicryle embedding is further described in a previous publication (*Collman et al., 2015*). Also, in Lowicryle, not all antibodies work as in LR White, specifically the VGluT2 antibody used in this study does not work in Lowicryle.

Immunohistochemistry is then carried out on the arrays using primary antibodies targeting the antigens of choice (Synapsin *Cell Signaling Technology* 5297S, VGluT2 *Millipore* AB2251, VGluT1 *Millipore* AB5905, Synaptophysin *Abcam* ab8049, PSD-95 *Cell Signaling* 3450, NR2B *NeuroMabs* 75–101, NR2A *Millipore* MAB5216, NR1 *Milipore* MAB363, GluR2 *Millipore* MAB397, EAAT2/GLT1 and EAAT1/GLAST gifts from Dr. Jeffrey Rothstein (Johns Hopkins Univeristy), GS *BD Biosciences* 610517, Gephyrin *BD Biosciences* 612632, GAD65 *Cell Signaling* 3988S, Gabra1 *NeuroMabs* 75–136, Bassoon *Abcam* ab13249, VGAT *SYSY* 131 011, FMRP *Dr. Justin Fallon* (Brown University), FXR1P *ProteinTech* 13194–1-AP, FXR2P *Dr. Rob Willemsen* and *DSHB* 1G2). The primary antibodies are visualized using fluorescently labeled secondary antibodies (Alexa 594, *Invitrogen* A11037, Alexa 488, *Invitrogen* A11034, and Alexa 647, *Invitrogen* A21245), and mounted in SlowFade Gold antifade with DAPI (*Invitrogen*).

### Data collection

Data for *Figures 1–5* and *Figures 1*, *3* and *5* supplements were newly collected from WT C57BL/6 Layer 4 and 5 somatosensory cortices in volumes of 140 μm × 100 μm x 5 μm). Data for *Figures 6* and *7* and supplemental *Figures 4* and *5* were generated from pooled data collected from six WT and six Fmr1KO mice that were imaged for a previous study (*Wang et al., 2014*). Fmr1KO mice were derived from Fmr1 KO2 animals originally from the FRAXS-sponsored distribution facility at Baylor College headed by Dr. David Nelson. Data volumes ranged from 140 × 100 μm x 0.5 μm to 140 × 100 μm x 2 μm. The antibody staining sequence for the tissue is: GAD 594 VGAT 488 VGluT2 647; Synpod 594 GEPH 488 VGluT1 647; FXR1 594, SYNPH 488; EAAT2 594 Bassoon 488; PSD95 594 GABAa1alpha 588; Synapsin 594 FMRP 488.

### Microscopy

Wide-field imaging of ribbons were accomplished on a *Zeiss* Axio Imager.Z1 Upright Fluorescence Microscope with motorized stage and Axiocam HR Digital Camera as previously described (*Micheva and Smith, 2007*). A position list was generated for each ribbon array of ultrathin sections using software modules for Axiovision. Single fields of view were imaged for each position in the position list using a *Zeiss* 63 ×/ 1.4 NA Plan Apochromat objective.

### Image registration and processing

Image stacks from ATD were imported into FIJI and aligned using both rigid and affine transformations with the Register Virtual Stacks plugin. The aligned image stacks were further registered across image sessions using FIJI and TrackEM.

The aligned and registered image stacks were imported into Matlab (*Mathworks*) and deconvolved using the native implementation of Richardson-Lucy deconvolution with empirical PSF through ten iterations (*Wang and Smith, 2012*). Custom functions were written to automate and facilitate this work flow (all custom Matlab functions are included in the publication). Empirical PSF were measured using 110 nm beads mounted on slides in the actual imaging system (*Wang and Smith, 2012*).

## Center of mass calculations for protein puncta

Matlab native function (regionprops) was used to calculate the centers of mass of puncta in the image volumes using 26 neighborhood 3D connected component analyses with an assumed background threshold that is 0.1 of the total dynamic range, which is 6553.5 for a 16-bit image, and is in line with previous background thresholds used for ATD analysis (*Wang and Smith, 2012*). Custom functions in Matlab were implemented to facilitate the handling and processing of the data (all custom Matlab functions are included in the publication).

## Custom software

Matlab software written for the synapse analysis, histogram generation and image deconvolution are included in the submission as a single archive titled 'Image Analysis Suite'.

## Resolution beyond 100 nm pixel resolution of the images

The pixel resolution of each single image used to generate the image volume is 100 nm. The section thickness is ~100 nm thus creating nearly 100 nm isotropic voxels. The resolutions in the distribution histograms are better than the 100 nm voxel resolution. This improved resolution arises from the statistics that combine tens of thousands of centers of mass calculations, which introduces sub-pixel jitter into the dataset which, over the large population, creates a dithering effect that provides sub-100 nm localization seen in our probability distributions.

## Discretization effects mediated by aliasing and ringing in 3D histograms

Halos are present in every distribution generated by SubSynMAP. These are caused by the discretization of the protein structures through pixilation. While the 3D weighted centroid calculations do provide sub-pixel coordinates, the very nature of the calculation being made on a discrete dataset increases the probability of certain sub pixel distances. The reason why these halos are more apparent in some instances is because the background is higher for those proteins; and the reason for a higher background is caused by: (1) fewer pooled synapses, some synapses are rarer and thus their histograms involve smaller protein numbers; and (2) some proteins have more background, because all protein centers within a 1 μm radius sphere are incorporated into the histogram. This means that even protein puncta not associated with that particular synapse can be included. As our synapses are aligned, the random non-synaptic signals will reduce to an even background (see randomized data), but the coherent signal (the protein localization within the aligned synapses) will rise from that homogenous noise.

## Synapse classification and rotations

The classification method has been published previously (*Wang et al., 2014*; *Allen et al., 2012*; *Hiu et al., 2016*). In short, classification occurs in a 1 μm radius sphere centered on the origin (PSD95 for excitatory synapses and gephyrin for inhibitory synapses). For example, a VGluT1 synapse will have PSD95 at the postsynaptic termini and synapsin at the presynaptic termini. Moreover, VGluT1 will be closer to synapsin than to PSD95. Thus, the algorithm will require the third point (VGluT1) to be closer to synapsin by a specified factor than it is to PSD95 in order for it to classify the grouping as a synapse. In some instances, after this initial classification step, a synaptic axis still retains more than one third point (VGluT1 puncta). If that is the case, then the algorithm looks through the entire dataset and removes sharing of the third point in a manner that maximizes the number of possible synaptic axis with third points. If a decision must be made regarding a single third point that is shared by two synaptic axes, then the axis that is the closest to the third point is selected. Finally, once the ownership of the third point has reached parsimony, a final check is applied to ensure that there is no duplication of the synaptic axis. This situation is rare in practice, but if it does occur, the axis with the smallest magnitude is preferred.

The output of the classified data is a set of synaptic axes defined by two points. These synaptic axes are then aligned. For the alignment of the synapses, the first step is to translate all of the synaptic axes so that the origin lies at 0, 0, 0 of the coordinate system. Then all the axes are aligned using a 3D rotation matrix as defined by:

$$R_x R_y R_z = \begin{bmatrix} \cos\alpha\cos\beta & -\cos\gamma\sin\alpha + \cos\alpha\sin\beta\sin\gamma & \cos\alpha\cos\gamma\sin\beta + \sin\alpha\sin\gamma & 0 \\ \cos\beta\sin\alpha & \cos\alpha\cos\gamma + \sin\alpha\sin\beta\sin\gamma & \cos\gamma\sin\alpha\sin\beta - \cos\alpha\sin\gamma & 0 \\ -\sin\beta & \cos\beta\sin\gamma & \cos\beta\cos\gamma & 0 \\ 0 & 0 & 0 & 1 \end{bmatrix}$$

where the rotation $R$ is rotated by $\alpha$ around the x-axis, $\beta$ around the y-axis, and $\gamma$ around the z-axis.

We rotate the synaptic axis to a position on the x-axis so that the coordinates of all the termini is x, 0, 0. We then use the rotation defined by this axis to rotate all associated third points for all channels in the dataset. For the distributional histograms, the dataset in which all of the protein centers across all imaged protein channels within a radius of 1 µm from the origin of the axis is used. This is done to ensure that the volume normalization is correct. The volume normalization assumes that there is some baseline probability that a punctum will co-localize with a synaptic axis, and that that probability increases as the volume increases, which is exactly what happens as you extend the distance from the origin to the termini of the axis. Thus, for the volume normalization to work properly, one must take all of the points scattered around the synaptic axis. This creates a 3D histogram, which can be rotationally reduced to 2D, in which the volumes of each bin are half slices of a spherical shell or a simpler approximation as a half slice of a cylindrical shell that can be described by the equation:

$$V = \frac{\pi h \left( r_1^2 - r_2^2 \right)}{2}$$

where $V$ = volume of the half cylindrical shell, $h$ = the y length of the bin, and $(r_1 - r_2)$ = the x length of the bin.

## Proteins associated with synaptic axis

An important note is that for every synaptic axis, all protein centers within a 1 µm radius sphere are incorporated into the histogram. This means that even protein puncta that are not associated with that particular synapse can be included. However, as our synapses are aligned, the random non-synaptic signals will reduce to an even background (see randomized data), but the coherent signal (the protein localization within the aligned synapses) will rise from that homogenous noise.

## Synapse histogram workflow

The synaptic vectors formed by connecting the centroids of PSD95 staining or of gephyrin staining and the centroids of staining for a vesicle protein were aligned in an x,y,z coordinate system. All PSD95 or gephyrin centroids were placed at the origin of the x,y,z coordinate system, and the centroids of a synaptic vesicle protein in each synapse were each placed at an appropriate position x, along the x-axis. Thus, the x axis is equivalent to the 'synaptic axis' in the coordinate system. The positions of other protein centroids within a 1 µm sphere of the origin of each synapse were rotated in the same way to appropriate positions in the x,y,z coordinate system. To provide a visual representation of the 2D distributions of these proteins in the x,y plane, with respect to the synaptic axis, the centroids of proteins were binned in the x,y plane as follows. The x coordinate of a centroid was given as its distance from the origin along the x-axis (*Figure 3A*). The y coordinate was given as its distance from the origin along the y-axis. The x coordinates of the centroids were binned within semi-circular bins of fixed width centered on the y-axis, and the y-coordinates were binned within semi-circular bins of the same fixed width centered on the x-axis. The semi-circular bins were mirrored along the x-axis (synaptic axis) to produce a circular image (*Figure 3B*). This creates a projection of centroids with x,y coordinates that are binned into circular bins of fixed width centered on the origin. To correct for the fact that the circular bins effectively draw from spherical bins in 3D, the number of centroids in each bin is divided by the volume of the corresponding spherical bin to give the density of centroids in each bin. This normalization ensures that the random probability of a centroid falling within each bin is equal.

## Chi-squared two-sample test

Generating a statistical comparison of both 2D and 1D non-parametric distributions was not trivial. We, with the help of the Stanford statistics department, settled upon the chi-square two-sample test (below) as the best solution. This is defined as follows:

$H_0$: The two samples come from a common distribution.

$H_a$: The two samples do not come from a common distribution.

Test statistic: For the chi-square two-sample test, the data are divided into $k$ bins and the test statistic is defined as:

$$\chi^2 = \sum_{i=1}^{k} \frac{(K_1 R_i - K_2 S_i)^2}{R_i + S_i}$$

Where the summation for bins 1 to $k$ , $R_i$ is the observed frequency for bin $i$ for sample 1, and $S_i$ is the observed frequency for bin $i$ for sample 2. $K_1$ and $K_2$ are scaling constants that are used to adjust for unequal sample sizes. Specifically,

$$K_1 = \sqrt{\frac{\sum_{i=1}^{k} S_i}{\sum_{i=1}^{k} R_i}}$$

$$K_2 = \sqrt{\frac{\sum_{i=1}^{k} R_i}{\sum_{i=1}^{k} S_i}}$$

## Sample size and replication

We used estimates derived from previous studies (*Wang et al., 2014*; *Allen et al., 2012*; *Hiu et al., 2016*), coupled with empirical tests to determine our sample size and replication. In this paper, synapses were pooled from six different animals per brain region (Layers 4 and 5) per genotype (WT and KO). Each animal was separately imaged and synapse classification was also done separately. These computationally generated synapses were then aligned and pooled to create average synapse protein data distributions.

## Synapse axes metrics

For 116592 Layer 5 cortical synapse axes: mean = 675.32 nm, median = 706.00 nm and standard deviation = 216.18. For 102886 Layer 4 cortical synapse axes: mean = 677.04, median = 704.50, standard deviation = 214.09. All these numbers are in line with published literature regarding the average size of a mouse cortical synapse (*Chen et al., 2008*; *Dani et al., 2010*; *Mineur et al., 2002*; *Chen et al., 2005*; *Tao-Cheng et al., 2014*; *Coleman et al., 2010*).

## Acknowledgements

We thank Drs. Michael Tranfaglia, Louis Leung and Logan Grosenick for helpful discussion and edits on the manuscript. We thank Drs. Yi Zuo and Xinzhu Yu for providing the FXS KO mice for the experiments. We thank Drs. Jeffrey D Rothstein, Justin Fallon, and Rob Willemsen for providing antibodies. Finally, we are grateful to the reviewers and editors for writing an excellent synopsis of our workflow, which is included in the 'Methods' under the heading 'Synapse histogram workflow'. We also thank Dr. Juan Cueva from *Aratome, LLC.* for providing the electron microscopy in this manuscript.

## Additional information

### Funding

| Funder | Grant reference number | Author |
| --- | --- | --- |
| John Merck Fund | | Gordon X Wang<br>Philippe Mourrain |
| FRAXA Research Foundation | | Gordon X Wang |
| National Institute of Neurolo- | 1R01NS092474 | Stephen J Smith |

| | | |
|---|---|---|
| gical Disorders and Stroke | | |
| National Institute of Mental Health | 1R01MH099647 | Philippe Mourrain |
| National Institute of Neurological Disorders and Stroke | 1R01NS062798 | Philippe Mourrain |
| National Institute of Diabetes and Digestive and Kidney Diseases | 1R01DK090065 | Philippe Mourrain |
| Allen Institute for Brain Science | | Stephen J Smith |

The funders had no role in study design, data collection and interpretation, or the decision to submit the work for publication.

## Author contributions

GXW, Conception and design, Acquisition of data, Analysis and interpretation of data, Drafting or revising the article; SJS, Conception and design, Drafting or revising the article; PM, Conception and design, Analysis and interpretation of data, Drafting or revising the article

## Author ORCIDs

Gordon X Wang, http://orcid.org/0000-0002-2707-7118

## Ethics

Animal experimentation: Animals were studied in accordance with animal use guide lines issued by the National Institutes of Health. All animals were handled with care in accordance with IACUC protocols at Stanford University. A minimum number of animal was used as necessitated by the experiments, and all animals were anesthetized using isofluorane to minimize suffering.

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
