## [Decision Letter]

Thank you for submitting your article "SUB-SYNaptic, Multiplexed Analysis of Proteins (SubSynMAP): Reveals FXR2P Localization deficits in *Fmr1* KO synapses" for consideration by *eLife*. Your article has been reviewed by three peer reviewers, one of whom is a member of our Board of Reviewing Editors, and the evaluation has been overseen by Eve Marder as the Senior Editor. The following individuals involved in review of your submission have agreed to reveal their identity: Shigeki Watanabe (Reviewer #3).

The reviewers have discussed the reviews with one another and the Reviewing Editor has drafted this decision to help you prepare a revised submission.

Summary:

The authors have developed a technique that maps the probability of protein distributions from array tomography data to infer the average localization of particular proteins over thousands of synaptic terminals. Array tomography uses conventional fluorescence microscopy but achieves super-resolution by physical sectioning of tissues, deconvolution and reconstruction of the arrays. Multiple proteins can be probed simply by repeating antibody staining against proteins of interest and the subsequent elution of the antibody. In brief, the new technique works as follows.

1) The synaptic vector is determined by connecting the centroid of fluorescence from a post-synaptic marker with the centroid of fluorescence from a vesicle marker. Another synaptic vesicle marker (synapse type specific) is used to ensure that the connected puncta represent a synapse. 2. The centroids of fluorescence puncta from other proteins are mapped relative to the synaptic vector, and histograms or density maps, are generated.

Using this method, the authors have demonstrated that: 1. Proteins with known localization are found at the appropriate locations relative to post-synaptic markers (PSD95 for glutamatergic neurons and gephyrin for GABAergic neurons). 2. FMRP is located mostly postsynaptically in Layer 5 neurons, but it is found both presynaptically and postynaptically in Layer 4 neurons. 3. These data are confirmed by mapping fluorescence data on underlying ultrastructure. 3. The FXR2P distribution, but not the FXR1P distribution, is altered in FMR1 KO mice via an unknown mechanism.

Essential revisions:

The reviewers felt that the method reported here as an extension of array tomography, and analysis methods already published is interesting and the data are compelling. It would be useful to researchers wanting to detect changes in protein distribution of synaptic proteins throughout brain regions caused by disease mutations, or other manipulations. However, for publication in *eLife*, the manuscript needs to be much more clearly written; it needs to provide more background information; and it needs to address several technical points.

The following major revisions will be necessary:

1) It is not made clear in the Abstract or Introduction what parts of the method have already been published and what parts are newly introduced in this manuscript. The authors should state in the Abstract that the method builds on data and analyses provided by ATD. Because the RL deconvolution is used here, and this was referred to as ATD in the authors' previous 2012 publication, the ATD terminology should be used here to avoid confusion. The newly introduced computational method supports alignment of all the synaptic vectors in such a way that the positions of additional proteins can be mapped and averaged over the thousands of synaptic axes identified in a sample.

2) The reviewers had quite a bit of confusion about the axes along which the projections to 2D were made. This is because the explanations in the Results and Materials and methods section are unclear and often ambiguous. The verbal descriptions of Figure 2, Figure 3 and Figure 4, their respective figure legends, and the work flow described in the methods section all must be thoroughly revised to make more clear the nature of the coordinate systems and the axes along which projections are made. After much re-reading, here is our understanding of what was done. The synaptic vectors formed by connecting the centroids of PSD-95 staining, or gephyrin staining and the centroids of staining for a vesicle protein were aligned in an x,y,z coordinate system. All PSD-95 or gephyrin centroids were placed at the origin of the x,y,z coordinate system, and the centroids of a synaptic vesicle protein in each synapse were each placed at an appropriate position x, along the x-axis. Thus, the x axis is equivalent to the "synaptic axis" in the coordinate system. The positions of other protein centroids within a 1 µm sphere of the origin of each synapse were rotated in the same way to appropriate positions in the x,y,z coordinate system. To provide a visual representation of the 2D distributions of these proteins in the x, y plane, with respect to the synaptic axis, the centroids of proteins were binned in the x-y plane as follows. The x coordinate of a centroid was given as its distance from the origin along the x-axis (Figure 3). The y coordinate was given as its distance from the origin along the y-axis. The x coordinates of the centroids were binned within semi-circular bins of fixed width centered on the y-axis, and the y-coordinates were binned within semi-circular bins of the same fixed width centered on the x-axis. The semi-circular bins were mirrored along the x-axis (synaptic axis) to produce a circular image (Figure 3). This creates a projection of centroids with x,y coordinates that are binned into circular bins of fixed width centered on the origin. To correct for the fact that the circular bins effectively draw from spherical bins in 3D, the number of centroids in each bin is divided by the volume of the corresponding spherical bin to give the density of centroids in each bin. This normalization ensures that the random probability of a centroid falling within each bin is equal. If the above description is not correct, the authors need to clarify the work flow further. In several figures, the density distributions are represented on a percentage scale, but the exact calculation by which the percentages were determined needs to be stated explicitly. The explanation of this process in a revised manuscript needs to be as clearly written as explanations of other methods were in earlier papers by the same authors.

3) The authors need to clarify and quantify the rate of false positives and negatives. The authors define a synapse by the presence of two synaptic vesicle markers close to the post-synaptic marker. However, this definition may not always be accurate. For example, in Figure 1, there is a pair of upper left signals containing juxtaposed PSD95 and vGlut1. However, this pair does not appear to be defined as a synapse because the SYN marker is not present. Since both vGlut1 and synapsin are vesicle markers, how do the authors explain this? Alternatively, the signals in the middle have all three markers, but the presynaptic markers are further away from the PSD95 marker. How well have the authors confirmed their definition of a synapse by the EM conjugate AT method? Is the pair of signals for PSD-95 and vGlut1 at the upper left, a false negative; that is a true synapse that has not been identified by the classification? Approximately what percentage of the synaptic vectors identified using this method actually represents synapses by morphology? If the algorithm used by the authors has been fully verified by the EM conjugate method in a previous publication, the authors need to state this in the Introduction, and give the calculated rate of false positive and false negative synapses identified by the algorithm. They need to summarize their previous studies with respect to choice of antibodies and the percentage of synapses that are actually labeled by the antibodies in the plastic sections. Is there a failure rate for labeling, for example, because of inadequate penetration of the antibody into the center of a section? A discussion of these issues, which may have been resolved in earlier papers, would be important for a tools and methods paper.

4) Supplemental Figure 1 is composed of portions of figures that were originally published in Wang and Smith, 2012 (PLoS Computational Biology 8(8): e1002671. doi:10.1371/journal.pcbi.1002671). That supplemental figure could be removed. If the authors feel strongly that it is necessary, all panels need to be properly attributed. The other supplemental figures must be associated with a regular figure, and appropriately named according to *eLife* instructions to authors.

5) The authors need to clarify and quantify the actual resolution of this method. The pixel dimensions are stated as 100 nm voxels in methods. Earlier papers indicate that an actual resolution of about 100 nm is achieved by the RL deconvolution. In the methods section, the authors indicate that statistics and dithering effect create a resolution "considerably better than 100 nm". The authors should discuss this issue more thoroughly in the Discussion section and provide an estimate of the actual resolution that is achievable. Is it ~10 nm? ~50 nm? Can this number be calculated?

6) It is difficult to assess the full extent of the 3D distribution of the signals from the 2D max projection. Please provide the 3D volumes as supplementary figures to Figure 3. The authors should also discuss how they exclude the possibility that the defining markers may be located outside of the reconstructed volume. For example, in Figure 1, would it be possible that the post-synaptic marker is actually outside the imaged volume (vice versa, the presynaptic markers are outside of the imaged volume)? This last question is related to the issue of false negatives.

7) There are some puzzling disparities in the data. The probability/ nm^3 is much higher in the 2D density plots than in the corresponding 1D distributions. Why? For example, in Figure 4, the 2D plot shows a 17% peak, but the 1D plot shows the peak is only ~3.3%. Please explain this.

8) In several of the 2D projections, there are periodic "halo" bins separated by bins with low density. These are particularly evident for bassoon and synaptophysin in Figure 5. What is the origin of these apparent artifacts? Could they be coming from inadvertent assigning of signals from neighboring synapses?

9) The criteria and accuracy of the EM conjugate AT experiment in Figure 6 need to be better explained. In Figure 6, there appear to be fluorescent signals that are not marked by arrows. For example, the middle panel in 6E shows both presynaptic and postsynaptic signals. The second panel from the left in 6F shows a second signal just outside of the synapses. What are the criteria that the authors use to decide which signals are associated with the particular synapse?

In the middle figure in 6F, the signal indicated by the arrow does not seem to be in the postsynaptic spine (the smear of the signal is in the post-synaptic terminal, but the bright spot is outside the boundary). In SubSynMAP, this signal will be counted into the postsynaptic bin; how would this affect the rate of false positives and negatives? (Note that the labels in Figure 6 all say L5, but should say L4. Please fix the typos.)

In general, the figure legend and description of this figure is not detailed enough. Are the fluorescence signals max projections of the same volume (210 nm in z) or 2.8 µm? If 2.8 µm, some of the signals observed may be from synapses located at a different location in z. The signals should be correlated from the same volume.

10) In Figure 7—figure supplement 2, Why is there no signal from vGlut2 in this condition?

---

## [Author Response]

*[…] Essential revisions:*

*The reviewers felt that the method reported here as an extension of array tomography, and analysis methods already published is interesting and the data are compelling. It would be useful to researchers wanting to detect changes in protein distribution of synaptic proteins throughout brain regions caused by disease mutations, or other manipulations. However, for publication in eLife, the manuscript needs to be much more clearly written; it needs to provide more background information; and it needs to address several technical points.*

*The following major revisions will be necessary:*

*1) It is not made clear in the Abstract or Introduction what parts of the method have already been published and what parts are newly introduced in this manuscript. The authors should state in the Abstract that the method builds on data and analyses provided by ATD. Because the RL deconvolution is used here, and this was referred to as ATD in the authors' previous 2012 publication, the ATD terminology should be used here to avoid confusion. The newly introduced computational method supports alignment of all the synaptic vectors in such a way that the positions of additional proteins can be mapped and averaged over the thousands of synaptic axes identified in a sample.*

We now mention that our method builds on ATD data in paragraph 1 of the Introduction. The novel and old parts of the method are more succinctly described in the first sentence of the second paragraph of the Introduction. ATD is now referenced throughout the text.

*2) The reviewers had quite a bit of confusion about the axes along which the projections to 2D were made. This is because the explanations in the Results and Materials and methods section are unclear and often ambiguous. The verbal descriptions of Figure 2, Figure 3 and Figure 4, their respective figure legends, and the work flow described in the methods section all must be thoroughly revised to make more clear the nature of the coordinate systems and the axes along which projections are made. After much re-reading, here is our understanding of what was done. The synaptic vectors formed by connecting the centroids of PSD-95 staining, or gephyrin staining and the centroids of staining for a vesicle protein were aligned in an x,y,z coordinate system. All PSD-95 or gephyrin centroids were placed at the origin of the x,y,z coordinate system, and the centroids of a synaptic vesicle protein in each synapse were each placed at an appropriate position x, along the x-axis. Thus, the x axis is equivalent to the "synaptic axis" in the coordinate system. The positions of other protein centroids within a 1 µm sphere of the origin of each synapse were rotated in the same way to appropriate positions in the x,y,z coordinate system. To provide a visual representation of the 2D distributions of these proteins in the x, y plane, with respect to the synaptic axis, the centroids of proteins were binned in the x-y plane as follows. The x coordinate of a centroid was given as its distance from the origin along the x-axis (Figure 3). The y coordinate was given as its distance from the origin along the y-axis. The x coordinates of the centroids were binned within semi-circular bins of fixed width centered on the y-axis, and the y-coordinates were binned within semi-circular bins of the same fixed width centered on the x-axis. The semi-circular bins were mirrored along the x-axis (synaptic axis) to produce a circular image (Figure 3). This creates a projection of centroids with x,y coordinates that are binned into circular bins of fixed width centered on the origin. To correct for the fact that the circular bins effectively draw from spherical bins in 3D, the number of centroids in each bin is divided by the volume of the corresponding spherical bin to give the density of centroids in each bin. This normalization ensures that the random probability of a centroid falling within each bin is equal. If the above description is not correct, the authors need to clarify the work flow further. In several figures, the density distributions are represented on a percentage scale, but the exact calculation by which the percentages were determined needs to be stated explicitly. The explanation of this process in a revised manuscript needs to be as clearly written as explanations of other methods were in earlier papers by the same authors.*

We found the description of the workflow above so compelling that we decided to include the italicized section above directly in our methods under the heading “Synapse histogram workflow”. We hope the reviewers and editors do not mind this. We have cited the reviewers and editors in the acknowledgement section for this. The Y axis scale is in percentage points in the figures because density is converted to percent probability by dividing the number of protein centers per bin by the number of synapses. This generates an estimated percent probability that a protein center can be found at a position in space in the average synapse of that class. This is stated in the text and is now reiterated in Figure 4 legend.

*3) The authors need to clarify and quantify the rate of false positives and negatives. The authors define a synapse by the presence of two synaptic vesicle markers close to the post-synaptic marker. However, this definition may not always be accurate. For example, in Figure 1, there is a pair of upper left signals containing juxtaposed PSD95 and vGlut1. However, this pair does not appear to be defined as a synapse because the SYN marker is not present. Since both vGlut1 and synapsin are vesicle markers, how do the authors explain this? Alternatively, the signals in the middle have all three markers, but the presynaptic markers are further away from the PSD95 marker. How well have the authors confirmed their definition of a synapse by the EM conjugate AT method? Is the pair of signals for PSD-95 and vGlut1 at the upper left, a false negative; that is a true synapse that has not been identified by the classification? Approximately what percentage of the synaptic vectors identified using this method actually represents synapses by morphology? If the algorithm used by the authors has been fully verified by the EM conjugate method in a previous publication, the authors need to state this in the Introduction, and give the calculated rate of false positive and false negative synapses identified by the algorithm. They need to summarize their previous studies with respect to choice of antibodies and the percentage of synapses that are actually labeled by the antibodies in the plastic sections. Is there a failure rate for labeling, for example, because of inadequate penetration of the antibody into the center of a section? A discussion of these issues, which may have been resolved in earlier papers, would be important for a tools and methods paper.*

The false positive and negative rate for the individual synapse classification is now included in the Introduction. A more detailed discussion of antibody testing and error rates is included in the methods. The error rate was first measured using computer to hand classification comparisons (Wang et al., 2014)[Supplemental Figure 1 – of that paper], and then reconfirmed using EM fiducials (Figure 8). In terms of marker colocalization, having looked at thousands of multiplexed images in 3D, it is not uncommon to find a VGluT1 and Synapsin colocalization without it being at a synapse. This due to the nature of the proteins where VGluT1 and Synapsin are more broadly expressed in neurons while a protein such as PSD95 are mostly localized to the synapse. This would biologically suggest two things, the site of production of these proteins (produced in the cell body and shipped to the synapses) and that biologically transport vesicles might harbor both proteins. The causes of false positive and negatives for the PSD95 and Synapsin antibodies used in this study have been explored using conjugate AT and EM (Collman et al., 2015)[Figures 9 and 10 – of that paper]. In short, false negative rates are due to failure of antibody detection, and false positives are due to circumstantial colocalization which occurs at low probability.

*4) Supplemental Figure 1 is composed of portions of figures that were originally published in Wang and Smith, 2012 (PLoS Computational Biology 8(8): e1002671. doi:10.1371/journal.pcbi.1002671). That supplemental figure could be removed. If the authors feel strongly that it is necessary, all panels need to be properly attributed. The other supplemental figures must be associated with a regular figure, and appropriately named according to eLife instructions to authors.*

The figure was originally included to explain certain aspects of ATD that was later removed for clarity. We are now removing this supplemental figure, as it is no longer necessary.

*5) The authors need to clarify and quantify the actual resolution of this method. The pixel dimensions are stated as 100 nm voxels in methods. Earlier papers indicate that an actual resolution of about 100 nm is achieved by the RL deconvolution. In the methods section, the authors indicate that statistics and dithering effect create a resolution "considerably better than 100 nm". The authors should discuss this issue more thoroughly in the Discussion section and provide an estimate of the actual resolution that is achievable. Is it ~10 nm? ~50 nm? Can this number be calculated?*

This is an important point and we have thought much about this. “Considerably better” was an overstatement, and it is toned down. We believe the idea of resolution and the interpretation of the data needed more discussion thus we added a new paragraph in the Discussion section to better explore the concepts. In short, in these distributions we are not resolving individual proteins at high resolution, as one would with EM, we are combining tens of thousands of observations that each individually have an amount of uncertainty attached (the original resolution is 100 nm^3^) but together provides an unequivocal localization map of that protein within the synapse. Thus, it would be difficult to place an exact number on the resolution, as traditional methods of visualizing a standard or ruler does not apply here. Thus, throughout the paper we have toned down the nanometer claims, e.g., first sentence of paragraph three in Discussion section “description of the nanometer substructure…” is now “description of the substructure”.

Author response image 1.EM verification of excitatory synapse classification fidelity.(**A**–**D**) Single large frame EM micrograph of mouse somatosensory cortex. (**A**) Raw PSD95 puncta from light level data superimposed on EM. (**B**) Hand traced synaptic clefts superimposed on EM. (**C**) Computer classified VGluT1 synapse post synaptic densities superimposed on EM. (**D**) Combination of **B** and **C**. (**A’**–**D’**) Zoomed in portions of the single EM micrograph. (**D’**) Black arrow point to a false positive and the white arrow point to a false negative. Note that the dataset is 3D thus some puncta not associated with clefts are associated in the z plane. Also due to the lack of a working VGluT2 antibody VGluT1 classified synapses are compared to all EM classified excitatory synapses, thus the false negative rate of VGluT1 is lower than the measured 12%.**DOI:**
http://dx.doi.org/10.7554/eLife.20560.021

*6) It is difficult to assess the full extent of the 3D distribution of the signals from the 2D max projection. Please provide the 3D volumes as supplementary figures to Figure 3. The authors should also discuss how they exclude the possibility that the defining markers may be located outside of the reconstructed volume. For example, in Figure 1, would it be possible that the post-synaptic marker is actually outside the imaged volume (vice versa, the presynaptic markers are outside of the imaged volume)? This last question is related to the issue of false negatives.*

We have generated 3D histograms with equal volume 10nm^3^ bins. The 3D histograms are displayed in Figure 3—figure supplement 1 and Video 1. As to the question regarding the post-synaptic marker being outside of the imaged volume, there are two answers. In terms of synapse classification, this is not possible, without post-synaptic PSD95 the algorithm will not classify a synapse. Thus, if there is a synapse with a PSD label outside of the search area then it will be a false negative that would be revealed in the above error analysis (Figure 8). However, outside of the definition of the synapse and the synaptic axis there is the possibility that a PSD protein lies outside of the 1um radius sphere we are focusing on. This would be a false negative in the sense that it won’t contribute to the 3D histogram, but in our tests using larger spheres, this does not contribute to the signal we see. For a more detailed description of signal and background see our answers to questions 8 and 9 below.

*7) There are some puzzling disparities in the data. The probability/ nm^3 is much higher in the 2D density plots than in the corresponding 1D distributions. Why? For example, in Figure 4, the 2D plot shows a 17% peak, but the 1D plot shows the peak is only ~3.3%. Please explain this.*

The reason is that the 1D bins are much bigger than the 2D bins. The 1D bins sweep a half halo through the 2D distribution thus it will cover dense 2D bins and less dense 2D bins. Thus the 1D bin is the average of those 2D bins, which can be lower.

*8) In several of the 2D projections, there are periodic "halo" bins separated by bins with low density. These are particularly evident for bassoon and synaptophysin in Figure 5. What is the origin of these apparent artifacts? Could they be coming from inadvertent assigning of signals from neighboring synapses?*

If you look carefully these halos are present in every distribution not just those in 5C. This is caused by the discretization of the signal cause by the pixels. While the 3D weighted centroid calculations do provide sub-pixel coordinates the very nature of the calculation being made on a discrete dataset increases the probability of certain sub pixel distances. The reason why these halos are more apparent in the inhibitory synapses is because the background is higher which is due to: 1) there are fewer inhibitory synapses, and 2) some proteins have more background. The cause of the background is described below in 9.

*9) The criteria and accuracy of the EM conjugate AT experiment in Figure 6 need to be better explained. In Figure 6, there appear to be fluorescent signals that are not marked by arrows. For example, the middle panel in 6E shows both presynaptic and postsynaptic signals. The second panel from the left in 6F shows a second signal just outside of the synapses. What are the criteria that the authors use to decide which signals are associated with the particular synapse?*

In fact, all protein centers within a 1um radius are considered for the histogram. We realize we did a poor job of describing this, and this actually informs the idea of background in the previous question (8). Thus, we have included the below section in the Materials and methods:

“Proteins associated with synaptic axis

An important note is that for every synaptic axis all protein centers within a 1um radius sphere is incorporated into the histogram. This means that even protein puncta not associated with that particular synapse can be included. However, since our synapses are aligned, the random non-synaptic signals will reduce to an even background (see randomized data), but the coherent signal (the protein localization within the aligned synapses) will rise from that homogenous noise.”

*In the middle figure in 6F, the signal indicated by the arrow does not seem to be in the postsynaptic spine (the smear of the signal is in the post-synaptic terminal, but the bright spot is outside the boundary). In SubSynMAP, this signal will be counted into the postsynaptic bin; how would this affect the rate of false positives and negatives? (Note that the labels in Figure 6 all say L5, but should say L4. Please fix the typos.)*

*In general, the figure legend and description of this figure is not detailed enough. Are the fluorescence signals max projections of the same volume (210 nm in z) or 2.8 µm? If 2.8 µm, some of the signals observed may be from synapses located at a different location in z. The signals should be correlated from the same volume.*

FMRP is not used for the classification of that synapse, thus it does not affect the error rates of synapse classification. That signal in 6F looks like it would in fact be segmented into two objects and they would contribute to both pre and post-synaptic bins. The error in Figure 6 is now fixed. The figure legend is also updated. The fluorescence signals are not max projections; they are aligned section by section with the EM.

*10) In Figure 7—figure supplement 2, Why is there no signal from vGlut2 in this condition?*

Unfortunately we could not find a VGluT2 antibody that work in lowicryle for conjugate AT/EM (This point is now noted in the methods). VGluT2 data at light level is from tissue embedded in LR White. This is why we could not assess the false positive and negative rates of VGluT2 synapse classification using EM, and why the false negative rate (Figure 8) for excitatory synapses is likely lower than the reported 12%.